# Occurrence of the Green Shield-Moss *Buxbaumia viridis* (Moug.) Brid. in the Bieszczady Mountains of Poland

**Piotr Brewczyński [1], Kamil Grałek [2] and Piotr Bilański [3],\***

1   Głogów Forest District, ul. Fabryczna 57, 36-060 Głogów Małopolski, Poland; Piotr.Brewczynski@krosno.lasy.gov.pl
2   The Regional Directorate of the State Forests in Krosno, ul. Bieszczadzka 2, 38-400 Krosno, Poland; Kamil.Gralek@krosno.lasy.gov.pl
3   Department of Forest Ecosystems Protection, University of Agriculture in Krakow, Al. 29 Listopada 46, 31-425 Krakow, Poland
\*   Correspondence: Piotr.Bilanski@urk.edu.pl; Tel.: +48-1266-253-69

**Abstract:** The small-sized gametophytes and sporophytes of the green shield-moss *Buxbaumia viridis* (Moug.) Brid. make it difficult to study. However, in Europe, there has been increasing interest in this species in the past few years, mostly as a result of the implementation of the Natura 2000 network. In Poland, *B. viridis* has only been reported in isolated studies that have been limited in terms of area and the number of participating workers. One of the Polish regions where *B. viridis* was recently recorded is the Bieszczady Mountains, but there have been no large-scale surveys of that region to date. The objective of the current work was to describe the *B. viridis* population in the Bieszczady Mountains in terms of its spatial distribution and abundance, investigate its selected microhabitat preferences, and evaluate the conservation status of this moss species within the Natura 2000 site Bieszczady PLC180001. The studied region encompassed 93,490.44 ha, including 69,056.23 ha of managed forests and 24,434.21 ha of forests belonging to the Bieszczady National Park. A preliminary survey was conducted in the Cisna Forest District (forest area of 19,555.82 ha) on 15–17 November 2017, while the main survey was performed in selected forest subcompartments of four forest districts—Baligród, Komańcza, Lutowiska, and Stuposiany—as well as the Bieszczady National Park from 5 to 16 November 2018. The field work consisted of searching for *B. viridis* sporophytes and setae and recording selected population and locality characteristics. The study led to the discovery of 353 new *B. viridis* localities in 202 study areas, with 9197 diploid individuals (sporophytes or setae only) growing in 545 microhabitats. The number of *B. viridis* localities discovered in the Bieszczady Mountains during 17 days of survey in 2017 and 2018 was two times higher than the combined number of localities previously found in Poland over more than 150 years (159 localities). Additionally, the number of sporophytes and setae identified was two times greater than their overall number in previous records. In addition, this study provides information about selected microhabitat preferences and the conservation status of this moss in the Bieszczady Natura 2000 site.

**Keywords:** *Buxbaumia viridis*; sporophyte; managed forests; national park; substrate; microhabitat; Natura 2000

## 1. Introduction

The green shield-moss *Buxbaumia viridis* (Moug.) Brid. is an epixylic (and sometimes terricolous) dioecious moss. The species is very difficult to study, as its microscopic gametophytes (<1 mm) are practically unobservable in the field. Its 1-2 cm-tall orthotropic sporophytes are not easily noticeable but can be found by trained workers. It is easier to spot young sporophytes in the autumn, when their capsules are yellowish-green and are clearly distinguishable from the substrate on which they grow. During snowy winters, it is practically impossible to find sporophytes, and they may also be damaged by snow cover. Thus, they are less likely to be discovered in the spring. When the capsules mature in the

springtime and turn dark green, yellow, and, finally, ochre-yellow, they become much more difficult to observe. Searching for *B. viridis* sporophytes is extremely time-consuming, as workers must meticulously explore potential microhabitats.

*Buxbaumia viridis* is an annual plant. Young individuals emerge in the autumn (October or November) and mature from March to June [1–3]. The number of spores per capsule is very large, much greater than that in other epixylic mosses, varying between 1.4 and 9.0 million, with a mean value of 6.0 million [4]. *Buxbaumia viridis* is found on deadwood, humus, and mineral soil on sites with high air humidity and on substrates with a relatively stable moisture content [5–7]. In Poland, *B. viridis* has been mostly reported on fir and spruce and less often on beech deadwood [1,5], with a few occurrences on humus or mineral soil [8,9]. In 2016, in the Białowieża Forest, *B. viridis* was first reported on oak deadwood (*Quercus robur* L.) with 78 sporophytes identified on two logs [10]. Due to its strong reliance on dead and decomposing wood, the species is strictly associated with forest habitats and occurs in a variety of forest communities: the Carpathian beech forest *Dentario glandulosae-Fagetum* (and especially its sub-community, the Carpathian fir forest *Dentario glandulosae-Fagetum abietetosum*), thermophilous fir forest (*Carici albae-Fagetum abietetosum*), thermophilous beech forest (*Carici albae-Fagetum typicum*), acidophilous beech forest (*Luzulo pilosae-Fagetum*), oak-lime-hornbeam forest (*Tilio-Carpinetum*), and oak-hornbeam forest (*Stellario-Carpinetum*) [1]. Generally, most *B. viridis* localities have been found in forests containing considerable proportions of silver fir *Abies alba* Mill. and Norway spruce *Picea abies* (L.) H. Karst. [2].

*Buxbaumia viridis* is found in North America [11], Europe [12,13], and Asia [14,15]. In Europe, it is known to be present in European Union (EU) countries such as Austria, Bulgaria, Croatia, Czech Republic, Denmark, Estonia, Finland, France, Germany, Greece, Hungary, Italy, Latvia, Lithuania, Poland, Romania, Slovakia, Slovenia, Spain, and Sweden [16], and it is also found in other countries such as Albania, Bosnia and Herzegovina, Montenegro, North Macedonia [17], Russia [18], Serbia [17], and the United Kingdom [7].

In Poland, to date, the largest *B. viridis* groups found in nature have consisted of several dozen sporophytes, while most authors have reported either individual sporophytes or groups comprising several individuals [1,5,9]. In 2014, the Polish population of *B. viridis* was estimated to contain approximately 4000 sporophytes [19]. Individual, scattered localities of this moss have been reported mostly in the Carpathians and Sudetes communities, as well as in the southern upland belt and the northern and western parts of the country [5]. In previous years, there has been increasing interest in *B. viridis*, mostly due to the establishment of the Natura 2000 network in Europe, which has led to the identification of numerous new localities of the moss including in Poland [19,20], Slovakia [1,21,22], Czechia [23], Hungary and other EU countries [16]. This phenomenon has been highlighted by Plášek [24], Chachuła and Vončina [25], Cykowska and Vončina [26], Vončina [1], Paciorek [21], and Kropik et al. [27]. These new findings have come from studies that were limited in terms of area and the number of workers engaged; however, they seem to suggest that this species may not be as rare in Poland as previously thought [28]. This is also consistent with information in the International Union for Conservation of Nature (IUCN) Red List of Threatened Species [29], where *B. viridis* is ranked as a species of least concern (LC) in Europe with a stable population trend. However, in 2001, in Poland, *B. viridis* was added to the list of protected species [30], probably due to the absence of comprehensive research on its occurrence across the country, especially in areas where it was previously reported to be in isolated, scattered localities. The focus on conservation areas (national parks and nature reserves) has led to a popular belief among researchers that *B. viridis* is much rarer in managed forests and that the likelihood of its survival there is low; therefore, special protection areas should be created around its localities in managed forests [1,2,19,20,31–34].

In Poland, *B. viridis* has been reported, relatively recently, to exist in the Bieszczady Mountains, where a total of 72 diploid individuals were found in five localities in 2010 [25]. In 2014, another 18 localities were recorded (16 in the Lutowiska Forest District and 2 in the

Stuposiany Forest District), with a total of 120 sporophytes (111 and 9, respectively) [19,20]. However, to date, no large-scale survey of *B. viridis* has been conducted in that region.

The objective of this work was to investigate the population of *B. viridis* in the Bieszczady Mountains and, especially, to determine its population size and distribution, assess the microhabitat preferences, and evaluate its conservation status in the Natura 2000 site Bieszczady PLC180001.

## 2. Materials and Methods

Studies on the occurrence of *B. viridis* in the Bieszczady Mountains were conducted as part of a periodical universal inventory of plants, animals, and other species and the parameterization of selected biotope characteristics in Poland by the State Forests National Forest Holding (PGL LP) pursuant to Director General decision no. 77 (16 April 2018). Work was conducted in forests managed by four Forest Districts—Baligród, Komańcza, Lutowiska, and Stuposiany—as well as in the Bieszczady National Park (BNP). The overall studied region was 93,490.44 ha (including 69,056.23 ha belonging to the Forest Districts and 24,434.21 ha belonging to the BNP). Due to the large size of the study area, the survey was conducted in selected forest subcompartments (forest areas that were homogeneous in terms of habitat conditions and major stand characteristics, as defined in a forest inventory). Each subcompartment constituted a separate study area. The subcompartments were selected as follows: The nationwide National Forest Inventory 4 × 4 km grid of circular sample plots [35] was extrapolated to a 1 × 1 km grid. The subcompartments containing the nodes of the grid were preliminarily qualified for the study. The grid for the Baligród, Cisna, and Komańcza Forest Districts is given in Figure 1.

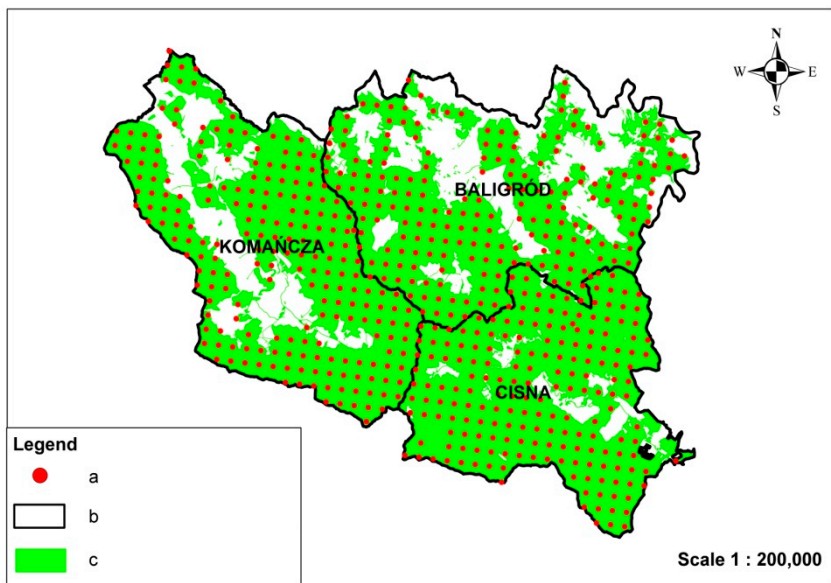

**Figure 1.** Nodes of a 1 × 1 km grid extrapolated from the 4 × 4 km grid of the National Forest Inventory: (**a**) centers of sample plots, (**b**) boundaries of Forest Districts, and (**c**) land belonging to State Forests Holding.

The initially qualified subcompartments were then checked to determine their species compositions and the ages of the tree stands contained in them to determine the likelihood of them having microhabitats suitable for *B. viridis*. The final qualifying criteria were as follows: stands with at least 20% of fir trees aged 70 years or more or the presence of alternative sapling stands with retained fir patches. For Stuposiany Forest District and the Bieszczady National Park, the percentage requirement for fir presence was lowered to 10% because of the insufficient number of subcompartments meeting the 20% criterion in these areas. A total of 250 forest subcompartments were selected; a detailed list of these is given below:

(1)    Baligród Forest District: 62 forest subcompartments, including 61 subcompartments with at least 20% of firs aged 70 years or more and one sapling subcompartment with some retained firs.
(2)    Komańcza Forest District: 46 forest subcompartments with at least 20% of firs aged 70 years or more.
(3)    Lutowiska Forest District: 64 forest subcompartments, including 60 subcompartments with at least 20% of firs aged 70 years or more and four sapling subcompartments with some retained firs.
(4)    Stuposiany Forest District: 40 forest subcompartments, including 19 subcompartment with at least 20% of firs aged 70 years or more, 17 subcompartments with at least 10% of firs aged 70 years or more, and four sapling subcompartments with some retained firs.
(5)    Bieszczady National Park: 38 forest subcompartments, including 19 subcompartments with at least 20% of firs aged 70 years or more and 19 subcompartments with at least 10% of firs aged 70 years or more.

The locations of the forest subcompartments selected for the *B. viridis* survey are given in Figure 2.

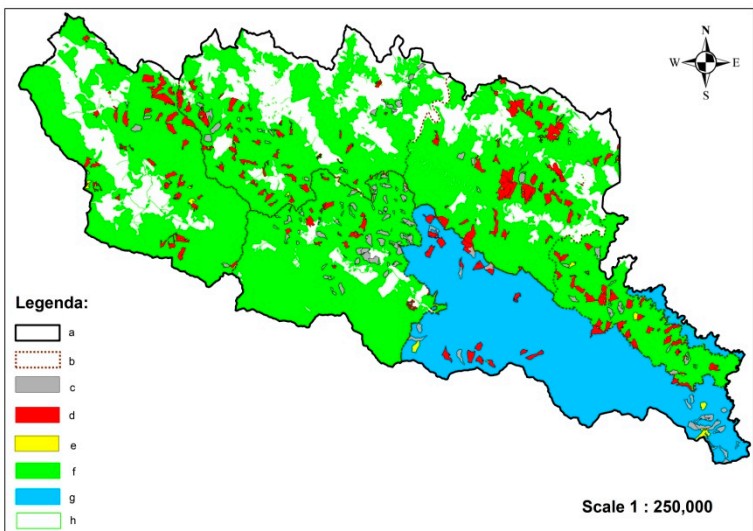

**Figure 2.** Distribution of study areas with *Buxbaumia viridis*: (**a**) boundaries of the study region, (**b**) boundaries of forest districts and the Bieszczady National Park (BNP), (**c**) surveyed areas without *B. viridis*, (**d**) selected study areas where *B. viridis* was found, (**e**) areas where *B. viridis* was found while moving to selected study areas, (**f**) land managed by forest districts, (**g**) BNP land, and (**h**) other forms of land ownership.

The field survey was conducted from 5 to 16 November 2018, by 30 workers with expert knowledge on *B. viridis*, divided into 10 teams consisting of three persons each. At the time of the survey, the studied region was free from snow cover. The teams surveyed the selected forest subcompartments focusing on fir, spruce, and beech deadwood, which have been reported to provide suitable microhabitats for the moss [1,5]. Due to the large sizes of the forest subcompartments in the mountains (the largest surveyed subcompartment was 82.29 ha in size), exploration work was concentrated in the vicinity of streams, where *B. viridis* sporophytes are most likely to occur [36]. While moving between streams, the workers surveyed potential microhabitats on the slopes away from water courses. It should be noted that despite the engagement of a large survey team, it was not possible to check all the potential microhabitats of the species within the study areas.

Upon finding *B. viridis* sporophytes, they were counted in their microhabitats and assigned to a locality. A locality was defined as either a single microhabitat (a deadwood fragment or patch of soil) with sporophytes or as several microhabitats if they were situated

within an area of approx. 25 m in diameter (five ares). Each new locality was immediately entered into a report form with the following data recorded: the geographic coordinates, subcompartment ID, number of sporophytes, number of logs occupied, number of sporophytes on each log, deadwood species (unless sporophytes occurred in the soil), dimensions of each log (length and diameter at midlength), and log decay stage (on a three-point scale: (1) slightly decomposed deadwood, (2) soft deadwood with a recognizable log shape, and (3) strongly decomposed, amorphous deadwood). The State Forest Information System was consulted to determine the type and year of the last silvicultural treatment for each subcompartment where *B. viridis* was found. It was assumed that no such treatments were performed within the borders of the Bieszczady National Park. Furthermore, the conservation status of each *B. viridis* locality was determined in accordance with the criteria contained in the Regulation of the Minister of the Environment of 30 March 2010, on preparing a conservation plan for a Natura 2000 site [37].

Each locality was photographed (each microhabitat with sporophytes and its surroundings at a given locality), and the photographs were stored in such a way so as to enable their identification in relation to a specific locality. Some localities were also found outside the study areas during the movement of the surveying teams between them. In those cases, only geographic coordinates were recorded.

On 15–17 November 2017, 33 workers grouped into 11 three-person teams conducted a pilot study in the Cisna Forest District (forest area of 19,555.82 ha). As that study was also aimed at finding the green fork tooth moss *Dicranum viride* (Sull. & Lesq.) Lindb., the aforementioned tree stand criteria were not applied. Study areas were defined as all subcompartments containing the nodes of a $1 \times 1$ km grid extrapolated from the $4 \times 4$ km grid of circular sample plots used for the National Forest Inventory (a total of 190 study areas). Due to the snowfall that occurred immediately before the survey, areas located in the southern part of the Forest District as well as those on north-facing slopes and at greater heights were excluded, as the snow cover there completely prevented *B. viridis* sporophyte observation. In total, 71 study areas were surveyed, with only 10 meeting the criteria adopted by the 2018 study. Due to the preliminary nature of that survey, the only data recorded were the number of sporophytes at a given locality, the number of microhabitats, and the geographic coordinates (data on the deadwood species, size, and decay stage were not collected). The conservation status of *B. viridis* at each locality was assessed according to the principles laid down in the Regulation of the Minister of the Environment of 30 March 2010, on preparing a conservation plan for a Natura 2000 site [37]. The State Forest Information System was consulted to determine the type and year of the last silvicultural treatment for each subcompartment where *B. viridis* was found. The preliminary data collected in the Cisna Forest District were used to determine the number of *B. viridis* localities, microhabitats, and sporophytes and the distribution of localities throughout the overall study area, as well as to evaluate the relationship between the occurrence of *B. viridis* and the type and date of silvicultural treatments and the conservation status of *B. viridis* in the Bieszczady Natura 2000 site. The other analyses were conducted without using the data obtained from the Cisna Forest District.

All the analyses and compilations (except for the number of sporophytes, localities, and their distribution in the study area) excluded the data collected during the movement of the survey teams between the selected forest subcompartments.

The $\chi^2$ test was performed to evaluate differences between proportions, and this was followed by use of the Marascuilo procedure for the pairwise comparison of proportions. These tests were carried out using the R software [38]. Other statistical analyses were conducted with Statistica Version 12 [39].

## 3. Results

The survey revealed 353 new green shield-moss localities in 202 study areas. A total of 9197 diploid individuals (sporophytes or setae) were found in 545 microhabitats. An additional 17 localities were identified when the survey teams were traveling to the study

areas. The exact number of sporophytes was counted for five of those localities (90 individuals), and in one case, the record read "several dozen sporophytes and setae". The number of sporophytes was not determined for the remaining additional localities (Supplementary Material Table S1).

The number of localities per forest district (or BNP) was 20 to 56, while the number of sporophytes ranged from 343 in the Cisna Forest District to 2467 in the Lutowiska Forest District (Table 1). The numbers of additional localities outside the designated study areas were as follows: four in the Baligród Forest District, four in the Komańcza Forest District, three in the Stuposiany Forest District, and six in the BNP. No locality was found in nature reserves (three study areas).

**Table 1.** Occurrence of *Buxbaumia viridis* in forest districts and in the Bieszczady National Park (BNP).

| Forest District or BNP | Study Areas N | Study Areas with *B. viridis* | | Localities N | Microhabitats N | Sporophytes and Setae N |
|---|---|---|---|---|---|---|
| | | N | % | | | |
| Baligród | 62 | 42 | 67.7 | 78 | 107 | 1936 |
| Cisna | 71 (10 [1]) | 14 (6 [1]) | 19.7 (42.9 [1]) | 14 | 38 | 343 |
| Komańcza | 46 | 41 | 89.1 | 74 | 127 | 1892 |
| Lutowiska | 64 | 56 | 87.5 | 104 | 161 | 2467 |
| Stuposiany | 40 | 29 | 72.5 | 55 | 79 | 1550 |
| BNP | 38 | 20 | 52.6 | 28 | 33 | 1009 |
| Others [2] | - | 14 | - | 17 | - | >90 |
| Total | 321 (275 [1]) | 202 (216 [3]) | 67.3 (78.5 [1]) | 353 (370 [3]) | 545 | 9197 (>9287 [3]) |

[1] Study areas meeting the criteria adopted in 2018. [2] *B. viridis* localities found while traveling to the selected study areas. [3] Total including *B. viridis* occurrences found while traveling to the selected study areas.

*Buxbaumia viridis* was found throughout the study region (Figure 2). The number of identified localities ranged from one to five per study area. *B. viridis* was found in 73.0% of the study areas in managed forests and in 52.6% of those in the BNP, but the difference failed to reach statistical significance ($\chi^2 = 1.9589$, df = 1, $p = 0.1616$). Managed forests contained 90.8% of the overall number of localities, 93.8% of the microhabitats, and 90% of the sporophytes identified, as compared to 9.2% of the localities, 6.2% of the microhabitats, and 10% of the sporophytes in the BNP. It should be noted that only 15.2% of all the study areas were located in the BNP.

*Buxbaumia viridis* occurred more often in study areas subjected to silvicultural practices (68.3%) than in those that had not been harvested for at least 15 years (or ever) (43.5%), with the difference being statistically significant ($\chi^2 = 14.2531$, df = 1, $p = 0.0002$). No statistically significant differences ($\chi^2 = 4.6822$, df = 2, $p = 0.0962$) were found between the frequency of *B. viridis* and the type of management procedures implemented (final felling, or intermediate or incidental cutting), although the highest mean number of localities was found in forest subcompartments harvested using the Swiss irregular shelterwood system. Those study areas also exhibited the greatest mean numbers of microhabitats and sporophytes. Again, those parameters were the lowest in areas where no cutting treatments had been implemented over at least the past 15 years (Table 2). No correlation between *B. viridis* occurrence and the time elapsed from the last procedure was identified (Figure 3).

The overall conservation status of *B. viridis* in the Natura 2000 site Bieszczady PLC180001 was found to be favourable (FV). All the parameters and indicators were also deemed to be favourable, except for the number of colonized logs, which was unfavourable-inadequate (U1). However, that indicator is not cardinal and affects neither the habitat parameter nor the overall conservation status (Table 3).

**Table 2.** Occurrence of *Buxbaumia viridis* depending on silvicultural treatment.

| Type of Treatment | Study Areas N | Study Areas with *B. viridis* N | *B. viridis* Localities N | *B. viridis* Microhabitats N | *B. viridis* Sporophytes and Setae N | Mean Number of *B. viridis* Localities per Study Area [1] | Mean Number of *B. viridis* Microhabitats per Study Area [2] | Mean Number of *B. viridis* Sporophytes per Study Area [3] |
|---|---|---|---|---|---|---|---|---|
| No treatment | 69 | 30 | 48 | 60 | 1201 | 1.60 [a] | 2.0 [b] | 40.03 [a] |
| Final felling | 99 | 66 | 130 | 208 | 3975 | 1.97 [a] | 3.15 [a] | 60.23 [a] |
| Incidental cutting | 112 | 83 | 139 | 221 | 3330 | 1.67 [a] | 2.66 [ab] | 40.12 [a] |
| Intermediate cutting | 41 | 23 | 36 | 56 | 691 | 1.57 [a] | 2.43 [ab] | 30.04 [a] |
| Total | 321 | 202 | 353 | 545 | 9197 | 1.75 | 2.70 | 45.53 |

[1,3] Within columns, values with the same letter are not significantly different ($p > 0.05$) according to the Kruskal-Wallis test and multiple comparisons of mean ranks for all groups of variable. [1] The Kruskal-Wallis test results were as follows: H (df = 3, $N$ = 202) = 4.819561, $p = 0.1855$; [3] The Kruskal-Wallis test results were as follows: H (df = 3, $N$ = 202) = 7.458503, $p = 0.0586$. [2] Within columns, values with different letters are statistically different ($p < 0.05$) according to the Kruskal-Wallis test and multiple comparisons of mean ranks for all groups of variable. The Kruskal-Wallis test results were as follows: H (df = 3, $N$ = 202) = 12.78195, $p = 0.0051$ (final felling vs. no treatment 0.008290).

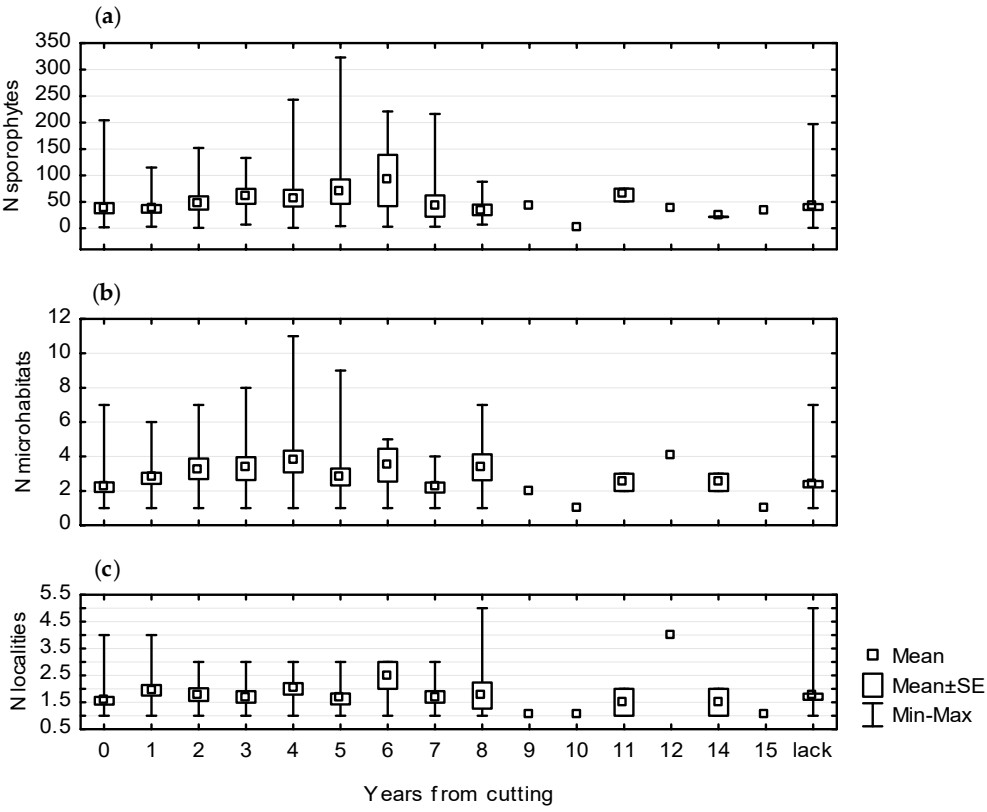

**Figure 3.** Relationship between the occurrence of *Buxbaumia viridis* and the time elapsed from the last silvicultural treatment. (**a**) Number of sporophytes as a function of years from cutting. The Kruskal-Wallis test results were as follows: H (df = 15, $N$ = 202) = 12.03537, $p = 0.6763$. (**b**) Number of microhabitats as a function of years from cutting. The Kruskal-Wallis test results were as follows: H (df = 15, $N$ = 202) = 17.21029, $p = 0.3065$. (**c**) Number of localities as a function of years from cutting. The Kruskal-Wallis test results were as follows: H (df = 15, $N$ = 202) = 14.95010, $p = 0.4550$.

**Table 3.** Evaluation of *Buxbaumia viridis* conservation in the Natura 2000 site Bieszczady PLC180001.

| Parameter [1]/Indicator ID and Name | | Cardinal Indicator | Shares of Localities of a Given Conservation Status [2] | | | | Overall Status [2] |
|---|---|---|---|---|---|---|---|
| | | | FV | U1 | U2 | XX | FV |
| I./1 | Number of sporophytes | Yes | 71.6% | 28.4% | 0.0% | 0.0% | FV |
| I./2 | Population area | No | 50.3% | 49.3% | 0.3% | 0.0% | FV |
| II./1 | Number of colonized logs | No | 1.4% | 98.6% | 0.0% | 0.0% | U1 |
| II./2 | Potential habitat area | Yes | 97.6% | 2.4% | 0.0% | 0.0% | FV |
| II./3 | Occupied habitat area | No | 87.0% | 13.0% | 0.0% | 0.0% | FV |
| II./4 | Habitat fragmentation | Yes | 93.2% | 6.8% | 0.0% | 0.0% | FV |
| II./5 | Shade | Yes | 59.6% | 40.4% | 0.0% | 0.0% | FV |
| II./6 | Air humidity | Yes | 53.8% | 42.5% | 3.8% | 0.0% | FV |
| II./7 | Tree and shrub cover | No | 86.0% | 14.0% | 0.0% | 0.0% | FV |
| II./8 | Ground vegetation cover | No | 99.0% | 1.0% | 0.0% | 0.0% | FV |
| II./9 | Bryophyte cover and characteristics | No | 94.2% | 5.1% | 0.7% | 0.0% | FV |
| II./10 | Competing moss species | No | 69.5% | 26.4% | 4.1% | 0.0% | FV |
| II./11 | Expansive species | No | 96.9% | 3.1% | 0.0% | 0.0% | FV |
| II./12 | Alien and invasive species | No | 100.0% | 0.0% | 0.0% | 0.0% | FV |
| III./1 | Conservation prospects | No | 99.7% | 0.3% | 0.0% | 0.0% | FV |

[1] Parameter: I. Population; II. Habitat; III. Likelihood of species survival. [2] Conservation status: FV—favourable, U1—unfavourable-inadequate, U2—unfavourable-bad, XX—unknown.

*Buxbaumia viridis* sporophytes were mostly found on deadwood, with a single exception where they grew directly on the soil. Throughout the study area, the predominant substrate for the moss was *Abies alba* deadwood (464 occupied logs). In the case of other types of deadwood (*Fagus sylvatica* L., *Salix* sp., *Picea abies*, *Populus tremula* L., and *Alnus incana* (L.) Moench.), the number of logs was less than 20 per species, with the greatest mean densities of *B. viridis* found on *Salix* sp. and *A. alba*, and the lowest on *A. incana*. The differences in occupation frequency between *A. alba* and *Salix* sp. logs and *A. incana* logs were statistically significant (Table 4).

**Table 4.** Occurrence of *Buxbaumia viridis* on different types of substrate.

| Type of Substrate | Sporophytes N | Microhabitats N | Mean Number of Sporophytes per Log [1] |
|---|---|---|---|
| *Abies alba* Mill. | 8362 | 464 | 18.02 [a] |
| *Fagus sylvatica* L. | 229 | 19 | 12.05 [ab] |
| *Salix* sp. | 120 | 5 | 24.00 [a] |
| *Picea abies* (L.) H. Karst. | 105 | 8 | 13.13 [ab] |
| *Populus tremula* L. | 25 | 4 | 6.25 [ab] |
| *Alnus incana* (L.) Moench. | 6 | 5 | 1.20 [b] |
| Soil | 5 | 1 | 5 [2] |
| No data | 345 | 39 | 8.85 [2] |

[1] Values marked with different letters differed significantly (*p* < 0.05) according to the Kruskal-Wallis test and multiple comparisons of mean ranks for all groups of variables. The Kruskal-Wallis test results were as follows: H (5, N = 505) = 13.33908, *p* = 0.0204. [2] Logs of undetermined species and single occurrences on the soil were excluded from the statistical analysis.

*Buxbaumia viridis* occurred with the greatest frequency on logs in Decay Stage 2 and sporadically on logs in Decay Stage 1. The number of sporophytes per log was also the greatest for Decay Stage 2 (it should be noted that those logs had the greatest area available for *B. viridis*). However, the greatest mean density of sporophytes (per 1 m$^2$ of log rather than per log) was found for deadwood in Decay Stage 3, with the difference being significant with respect to Decay Stage 2 (Table 5).

**Table 5.** Occurrence of *Buxbaumia viridis* depending on the decay stage of logs.

| Deadwood Decay Stage | Sporophytes N | Logs N | Mean Log Area m$^2$ | Mean Number of Sporophytes per Log | Mean Number of Sporophytes per m$^2$ of Log [1] |
|---|---|---|---|---|---|
| 1 | 20 | 6 | 3.4 | 3.3 | 4.0 |
| 2 | 7149 | 379 | 7.2 | 18.9 | 12.9 [a] |
| 3 | 1611 | 116 | 4.0 | 13.9 | 14.5 [b] |
| No data | 412 | 43 | 1.7 | 9.6 | 7.9 |

[1] Values marked with different letters differed significantly (*p* < 0.05) according to the Mann-Whitney U test (N = 538) = 19123.5, *p* = 0.033995.

The largest number of occupied microhabitats belonged to the diameter class 7–30 cm and the length class >3 m, although some sporophytes were also found on pieces <7 cm in length (Table 6).

**Table 6.** Number of logs occupied by *Buxbaumia viridis* by length and diameter class.

| Length Class | Diameter Class | | | | | Total |
|---|---|---|---|---|---|---|
| | **<7 cm** | **7–30 cm** | **30–50 cm** | **≥50 cm** | **No Data** | |
| <0.5 m | 6 | 8 | | 2 | | 16 |
| 0.5–1 m | | 19 | 4 | 2 | | 25 |
| 1–3 m | 2 | 93 | 38 | 15 | | 148 |
| >3 m | | 175 | 95 | 46 | | 316 |
| No data | | | | | 39 | 39 |
| Total | 8 | 295 | 137 | 65 | 39 | 544 |

As log dimensions determine the areas of potential habitats, the density of sporophytes per 1 m² of microhabitat seems to be an indicator that better reflects the preferences of *B. viridis* with respect to the deadwood size. The data analysis revealed that the areas of greatest sporophyte density occurred on thin and short deadwood fragments, and the densities in these areas were significantly higher than those on the thickest and longest logs (Figure 4).

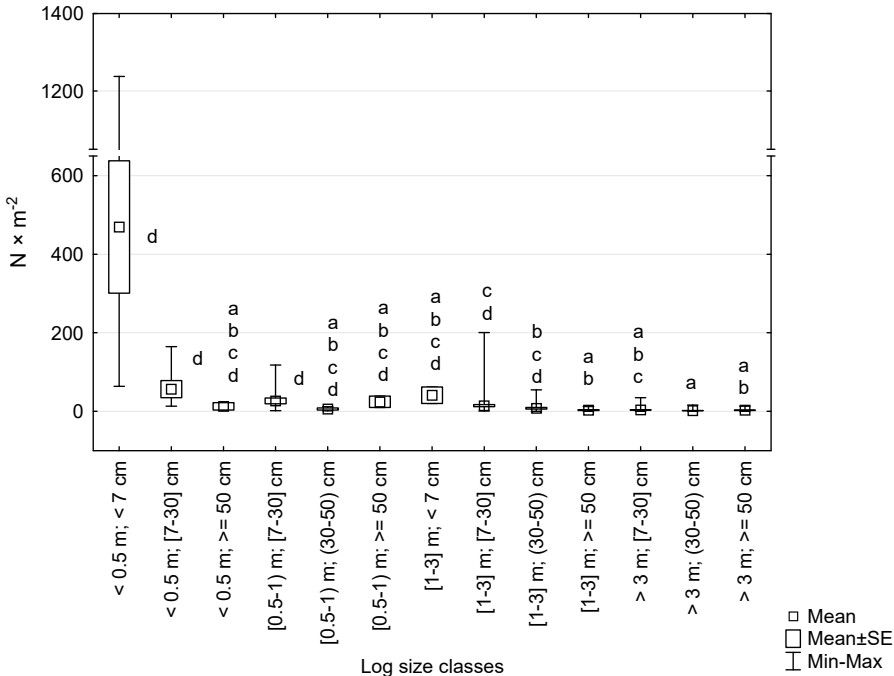

**Figure 4.** Sporophyte density on logs depending on their size classes (length, m; diameter, cm). Values designated by different letters differed significantly (*p* < 0.05) according to the Kruskal-Wallis test and multiple comparisons of the mean ranks for all groups of variables. The Kruskal-Wallis test results were as follows: H (12, *N* = 505) = 166.7670, *p* < 0.001.

The greatest numbers of logs with *B. viridis* sporophytes were found in localities with very high levels of air humidity, especially near streams and bog springs. Many sporophytes also occurred in localities situated away from water courses, but not in dry areas (e.g., north- and west-facing slopes), with the mean number of individuals per log being almost the same as in areas with high and very high air humidity conditions. While some *B. viridis* localities were also observed in dry places (e.g., south- and east-facing slopes, near mountain peaks), these occurrences were sporadic with a markedly lower

density of sporophytes as compared to moist localities, although differences in terms of density were not statistically significant (Table 7).

**Table 7.** Occurrences of *Buxbaumia viridis* depending on air humidity at the locality.

| Air Humidity Level | Microhabitats N | Sporophytes N | Mean Number of Sporophytes per Microhabitat [1] |
|---|---|---|---|
| Very humid | 313 | 5353 | 17.10 [a] |
| Humid | 216 | 3703 | 17.14 [a] |
| Dry | 16 | 141 | 8.81 [a] |

[1] Values with the same letter are not significantly different ($p > 0.05$) according to the Kruskal-Wallis test and multiple comparisons of mean ranks for all groups of variable. The Kruskal-Wallis test results were as follows: H (df = 2, $N = 545$) = 2.915299, $p = 0.2328$.

## 4. Discussion

Prior to the present study, only 23 isolated, scattered *B. viridis* localities had been reported in the Bieszczady Mountains; however, the current findings show that the species is widespread in that region. The number of *B. viridis* localities recorded during a 3-day pilot survey in 2017 and a 2-week comprehensive survey in 2018 was twice as high as the overall number of localities identified throughout Poland over the past 150 years (159), and the same was true for the number of sporophytes and setae [19,20]. Most of the previously found localities contained several sporophytes or single individuals, with the largest groups containing several dozen sporophytes [1,5,9]. In this study, we found 18 groups consisting of over 100 sporophytes per microhabitat (the largest group contained 262 sporophytes) and 33 groups consisting of 50 to 100 sporophytes. Although *B. viridis* is difficult to observe, during the survey, it was not only found in the selected study areas but also when workers were moving between the areas. These results suggest that, in other Polish and European regions, *B. viridis* could be more widespread than previously thought based on findings from isolated localities [23,40]. By the same token, the viewpoint that *B. viridis* is extremely rare [1,2,5,9,19,20,26,41] has been proven wrong. Similar hypotheses were previously made by, amongst others, Kozik and Vončina [28] and Hajek [9,41], who observed increases in *B. viridis* occurrences after the implementation of the Natura 2000 network, when the species attracted researchers' attention as a subject of protection under Natura 2000. To verify this hypothesis, however, it is necessary to perform comprehensive surveys, similar to that conducted in the Bieszczady Mountains, in other regions in which isolated *B. viridis* localities have been reported in recent years. If it turns out that the species is as abundant in other Polish and European regions as in the Bieszczady Mountains, then *B. viridis* should be deleted from the list of protected species in Poland [42], the European "Red list of mosses" [43], Annex I to the Bern [44], and Annex II to the Habitats [45], as the species was entered into those documents based on its rarity and the fact that half of its localities had been identified a very long time ago [2].

The concentration of previous surveys on protected areas (national parks and nature reserves) has led to the false belief that *B. viridis* occurs in stands subjected to rational forest management much less often and that its survival in those habitats is threatened [1,2,19,20,31–33,46]. Szmajda et al. [5] went so far as to claim that the species clearly avoids places affected by human activity. However, the current findings do not confirm any of those assertions. Indeed, in managed forests, *B. viridis* was found more often than in the BNP (although the difference was not statistically significant). At the same time, it should be noted that, while the amount of deadwood was high throughout the study region, in managed forests (nearly 52 m$^3$/ha) [47], it was somewhat lower than in the BNP (58 m$^3$/ha) [48]. The theory about the extreme susceptibility of *B. viridis* to human activity appears even more conspicuously false when one analyzes the relationship between moss occurrence and silvicultural treatments. *B. viridis* was more often found in areas with a recent history of management practices than in areas that had not undergone treatment for at least 15 years (or ever), with the difference being statistically significant. This is consistent

with the findings of Holá et al. [23], Gawryś and Szulc [49], and Deme et al. [40] and contradicts the view that only protected areas (nature reserves and national parks such as the BNP) provide suitable habitats for the studied moss species [25] and that strict legal protection is the best way of maintaining *B. viridis* localities [1]. Thus, calls for creating protective zones around *B. viridis* localities in managed forests are unsubstantiated [1,10,19,20,31,34,50]. It should be expected that, in addition to the presence of its preferred substrates, the occurrence of *B. viridis* is determined by factors other than silvicultural procedures, e.g., air humidity [28] and substrate pH or nitrogen/phosphorus proportions [51]. However, the identification of these factors requires further detailed study.

To date, in Poland, *B. viridis* has been mostly reported to occur on spruce and fir deadwood, less often on beech deadwood [1,5], and only occasionally on humus or mineral soil [8,9]. In 2016, *B. viridis* was reported for the first time on oak deadwood in the Białowieża Forest, with 78 sporophytes growing on two logs [10]. The current study confirmed that the most suitable substrate for *B. viridis* is *A. alba* (its presence was a prerequisite for the selection of survey areas). Fir deadwood accounted for 85% of the *B. viridis* microhabitats and accommodated 90% of the sporophytes identified. The small proportion of *P. abies* deadwood occupied by *B. viridis* may be attributable to its small share in the tree species composition of the Bieszczady forests. The frequency and density of *B. viridis* on *F. sylvatica* deadwood was also very small (3.5% and 2.5%, respectively), despite the high share of that tree species in the overall pool of deadwood, which indicates that *F. sylvatica* is a less suitable substrate for *B. viridis*. The survey conducted in the Bieszczady Mountains also revealed that *B. viridis* sporophytes can grow in the soil (humus), although it is difficult to assess their frequency on that substrate due to the adopted methodology (examining deadwood) and the studied area. This is consistent with a Hungarian study that showed that soil may play a greater role in the presence of *B. viridis* than previously thought [40]. The current study revealed the presence of *B. viridis* on the deadwood of *Salix* sp., *Populus tremula*, and *Alnus incana* for the first time. It should be noted that the deadwood of those species was not the focus of the survey, as according to the adopted methodology, the study concentrated on the downed deadwood of *A. alba*, *F. sylvatica*, and *P. abies*, which are indicated in the related body of literature as being suitable habitats for *B. viridis* [1,5]. While it is impossible to draw definitive conclusions about the suitability of the aforementioned species of deadwood for *B. viridis* growth based on isolated findings, future studies should encompass other tree species and the soil, as the studied moss does not appear to be stenotopic in terms of substrate. It is conceivable that it may occur on a wide range of deadwood species found in Polish forests. Of particular interest is *Salix* sp. deadwood, which was found to exhibit the highest mean sporophyte density of all the studied species. This issue requires further comprehensive investigation.

The conservation status of *B. viridis* in the Natura 2000 site Bieszczady PLC180001 was deemed to be favourable (FV), with only one indicator of the "habitat" parameter falling in the inadequate category (U1). It seems that the criterion for that particular indicator is too stringent, as a favourable status would require the presence of six or more microhabitats per locality, defined as a circular area with a diameter of 25 m centered at the first site at which the moss was recorded. Due to the considerable dispersion of deadwood (substrate for *B. viridis*), it was difficult for this criterion to be met even in the Bieszczady Mountains, which have a deadwood density of more than 50 m$^3$/ha, so only 3% of the *B. viridis* localities qualified. However, this is not a cardinal indicator, and so it does not affect the evaluation of parameters in situations where the other indicators are deemed to be FV. It should be emphasized that 70% to 100% of the localities revealed FV evaluations for the other indicators. Furthermore, the FV conservation status was found in a region without a conservation plan or dedicated measures implemented for *B. viridis*. The share of managed forests subjected to silvicultural treatments in the Natura 2000 site Bieszczady was 64%. The conservation status of *B. viridis* in the managed forests was also found to be FV and did not differ from that for the overall area, which further proves that forest management does not adversely affect the *B. viridis* population, contradicting the theory

that the most suitable habitat conditions for that species can only be found in national parks and nature reserves [2]. These findings also undermine the view that the long-term survival of the species (parameter III "likelihood of species survival") depends on the form of protection, and FV should only be awarded to localities in national parks, nature reserves, or Natura 2000 sites under strict legal protection [1]. It should be added that such a view is inconsistent with the criteria set forth in the Regulation of the Minister of the Environment of 30 March 2010, for preparing a conservation plan for a Natura 2000 site [37]. Pursuant to that document, parameter III should be evaluated not in terms of the implemented forms of nature protection at *B. viridis* localities but in terms of actual threats to the species. Thus, the "likelihood of species survival" should be deemed favourable when the preservation of the species over a period of 10–20 years is almost certain without the need to undertake special measures to counteract the identified threats, as is the case in the Bieszczady Mountains.

According to Vončina and Chachuła [2], the presence of *B. viridis* does not depend on the deadwood diameter, with suitable conditions provided both by logs 40–50 cm in diameter and those with a diameter of only a few centimeters. Ellis et al. [52] reported the presence of *B. viridis* on a *P. abies* log 0.8 m in length. Cykowska [46] noted the occurrence of a sporophyte on a small piece of wood and a spruce stump. In a study by Hajek [9], *B. viridis* was found to prefer epixylic substrates, especially stumps. In the present survey, *B. viridis* was shown to grow on logs of various sizes, ranging from thin and short to very thick and long. Obviously, log size translates into the area of available habitat, as longer and thicker logs may accommodate more sporophytes. Thus, to determine the actual *B. viridis* preferences in terms of deadwood size, it was necessary to determine the species' density—the number of sporophytes per unit area of habitat ($m^2$). The greatest sporophyte density values were obtained for the thinnest and shortest deadwood fragments, and the values decreased with increasing deadwood dimensions, even though workers naturally tend to examine larger deadwood fragments, as they are more readily noticeable. Since the sporophyte density was the highest for thin and short materials, one should pay particular attention to those microhabitats during future comprehensive *B. viridis* distribution surveys, as that could increase the detectability of the species.

Previous work has shown that *B. viridis* grows on deadwood devoid of bark, on decaying stumps and downed decomposing logs, and sometimes also on humus-rich soil [1,5,9]. To date, few studies have analyzed the degree of decomposition of deadwood occupied by *B. viridis*. In a study by Hajek [9], *B. viridis* was found to occur predominantly on strongly or very strongly decayed deadwood: on a six-point scale, most microhabitats were found to be in Decay Stages 5 and 6 (defined as blade penetration of 51–100 mm and >100 mm, respectively). Holá et al. (2014) reported that most microhabitats were associated with Decay Stage 6 on an eight-point scale (fragments of soft wood with a recognizable outline). In a study conducted in the Pieniny National Park by Vončina and Chachuła [2], *B. viridis* was found to prefer biotopes corresponding to Decay Stage 2 on a three-point scale (1—slightly decomposed deadwood; 2—intermediate stage; 3—advanced decomposition), where two-thirds of individuals were found. The observations made in the Bieszczady Mountains are consistent with those detailed in previous records, as *B. viridis* was most frequently identified on soft deadwood with a recognizable log outline (Decay Stage 2). To control for habitat size, the present study also used density (the number of sporophytes per $m^2$ of habitat), which was the highest for deadwood in Decay Stage 3. This is in line with reports indicating *B. viridis* prefers highly decomposed wood. The next stage of decomposition is humus, but it should be noted that under field conditions, it is often difficult to make a clear distinction between deadwood in an advanced decay stage and humus. The preference of *B. viridis* for highly decomposed deadwood indicates that the presence of its sporophytes on the soil (humus) might be more frequent than that suggested by the isolated case found in this study, especially because it is easier to identify potential *B. viridis* microhabitats on deadwood than in soil. This issue requires further study.

## 5. Conclusions

Rational forest management does not adversely affect the occurrence of *B. viridis*, as in this study, it was found to be more abundant in managed study areas than in the National Park. No correlation was found between the presence of *B. viridis* and the time from the last silvicultural treatment. The fact that *B. viridis* was found more often in managed forests than in the BNP contradicts the theory that only conservation areas provide suitable conditions for the growth of that species. The conservation status of *B. viridis* was favourable (FV) both for the entire Natura 2000 site Bieszczady PLC180001 and for the managed forests that it contains.

*Buxbaumia viridis* was most frequently identified on logs in Decay Stage 2, which also featured the greatest numbers of its sporophytes. The present study revealed the presence of *B. viridis* on logs of different dimensions, ranging from thin and short to very thick and long. The greatest *B. viridis* density was recorded on the thinnest and shortest deadwood fragments; its density decreased with increasing deadwood diameter and length.

The present study confirmed that *A. alba* deadwood is the most preferred substrate for *B. viridis*, while it was also recorded for the first time on deadwood of *Salix* sp., *P. tremula*, and *A. incana*. In one case, the sporophytes of this moss were found to grow directly into the soil (humus).

These findings indicate that *B. viridis* is widespread in the Bieszczady Mountains, contrary to what was expected based on previous reports concerning that area. This may also be the case for other regions of the country and Europe, where only isolated *B. viridis* localities have been recorded previously. This would mean that the population and number of localities of green shield-moss are currently underestimated.

**Supplementary Materials:** The following are available online at https://www.mdpi.com/1999-4907/12/3/374/s1. Table S1: New *Buxbaumia viridis* localities in the Bieszczady Mountains.

**Author Contributions:** Conceptualization, P.B. (Piotr Brewczyński) and K.G.; investigation, P.B. (Piotr Brewczyński), K.G., and P.B. (Piotr Bilański); methodology, P.B. (Piotr Brewczyński), K.G., and P.B. (Piotr Bilański); software, P.B. (Piotr Bilański); supervision, P.B. (Piotr Brewczyński) and P.B. (Piotr Bilański); validation, P.B. (Piotr Brewczyński), K.G., and P.B. (Piotr Bilański); visualization, K.G. and P.B. (Piotr Bilański); writing—original draft, P.B. (Piotr Brewczyński), K.G., and P.B. (Piotr Bilański); writing—review and editing, P.B. (Piotr Brewczyński) and P.B. (Piotr Bilański). All authors have read and agreed to the published version of the manuscript.

**Funding:** This research was funded by the State Forests National Forest Holding and partly by the Ministry of Science and Higher Education of the Republic of Poland.

**Institutional Review Board Statement:** Not applicable.

**Informed Consent Statement:** Not applicable.

**Data Availability Statement:** The data presented in this study are available in supplementary material.

**Acknowledgments:** The authors would like to express their gratitude to the following people who carried out the survey: Monika Armata, Patryk Babula, Zbigniew Boczar, Katarzyna Borowiec, Mateusz Czech, Radosław Czechyra, Marcin Czereczon, Zofia Daszkiewicz-Siuciak, Jarosław Gosztyła, Bartosz Janc, Łukasz Kłeczek, Joanna Kobeszko, Marceli Kot, Tomasz Kowszyński, Marek Krajnik, Kamil Kubacki, Alicja Liszkowska, Gaweł Liszkowski, Daria Majkowska, Tadeusz Maksymowicz, Kinga Mazur, Marek Morajko, Adrianna Natanek, Jacek Pasionek, Witold Pilch, Piotr Pyzia, Michał Rąpła, Marcin Scelina, Marek Sęk, Grzegorz Szafran, Sebastian Wójtowicz, Paulina Zdeb and Dariusz Żurek.

**Conflicts of Interest:** The authors declare no conflict of interest.

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
