# Peer review of "Occurrence of the Green Shield-Moss Buxbaumia viridis (Moug.) Brid. in the Bieszczady Mountains of Poland"

_forests, doi:10.3390/f12030374_

Round 1

Reviewer 1 Report

The work is very interesting. The research deepens the knowledge on the distribution of Buxbaumia viridis and on the threats to its habitat through a capillary and systematic study on its presence in the territory. I recommend a more careful revision of the English language.

The research is very interesting and makes an original and significant contribution to the study of the distribution and ecology of the species: fundamental aspects for its in situ conservation.The research follows on from the research conducted in other countries and helps to outline the distribution of the species in Europe.The contribution is very clear and the methodology adopted is rigorous and effective.The conclusions are also well developed and consistent with the aims of the research and therefore respond very well to the initial project.

Author Response

We would like to thank Reviewer 1 for their positive opinion and, as suggested, we performed a linguistic correction of the text. It has been carried out by MDPI's English editing service. Please see the attachment.

Reviewer 2 Report

The manuscript regarding the occurrence of B. viridis in the Bieszczady Mountains has been made on the basis of a huge sampling effort, with a well-designed methodology, and provides interesting and unexpected results to characterize the ecological preferences of this unnoticeable moss.

In my opinion, the manuscript has been written in a too local scope, and could be improved if the introduction and conclusions are moderately re-written. In the introduction, for example in lines 50-54, the characterization of B. viridis, is based on references of local studies, although the paragraph is written as a general characterization. I would suggest to confirm the provided data with the ecological preferences of this moss worldwide, and include the appropriate references. The same can be applied to the sentence in lines 66-67. Similarly, I think that a general description of the distribution of this species worldwide would be desirable in lines 71-76. I would recommend the authors to check the distribution of the species in the Natura 2000 net at: https://natura2000.eea.europa.eu/#

In the same line, I think that the discussion and conclusions could be improved if they are slightly re-written with a more general scope. I think that some of the results of the authors are of general interest to characterize and understand the real distribution of this moss (such as their findings on new types of wood), but this can only be addressed if the current discussion is completed with the knowledge on this moss worldwide.

I would like to point out one final concern regarding the conclusions of this paper. There are some parts of the manuscript (such as lines 325-326 or lines 332-337) where the authors suggest that according to their results this moss could be deleted from the lists of protected species. I think that the authors should be more careful on this statement. The Natura 2000 initiative aims to protect habitats that are endangered or rare at a European scale. Buxbaumia viridis distribution is linked to fir forests (as the authors confirm for example in lines 363-366), which is an undoubtedly rare type of forest when the whole Europe is considered. So, in the same line as my previous suggestions, I would avoid to conclude that B. viridis should not be included in protection lists at the European scale when its distribution is not studied at this global scale. Even though Poland and the surrounding territories can provide suitable environments that favour the development of this species, the authors cannot forget the larger scope and aim of the Nature 2000 initiative.

Minor mistakes:

  • Please check the expression “An additional 17 localities” (lines 204-205)
  • Figure 2 legend (lines 228-229): I would change the explanation of “c” to something like "surveyed areas without viridis findings"
  • Tables 2, 4 and 5 legends: please, explain what do “a” and “b” superscripts mean. Current expression is not clear. Which is the difference on significance among the two letters?
  • Figure 4 legend: this is not clear, which is the difference between a and b?
  • Table 7 legend: same comment as above. And be aware that there is only letter “a” here, the legend cannot say “different letters”
  • Discussion: please delete lines 308-3011
  • Line 437, use italics for viridis
  • Conclusions: please delete lines 458-459

Author Response

We thank Reviewer 2 for their constructive feedback. We performed a linguistic correction of the text with the help of MDPI. Please see the attachment.

Question 1. In my opinion, the manuscript has been written in a too local scope, and could be improved if the introduction and conclusions are moderately re-written. In the introduction, for example in lines 50-54, the characterization of B. viridis, is based on references of local studies, although the paragraph is written as a general characterization. I would suggest to confirm the provided data with the ecological preferences of this moss worldwide, and include the appropriate references.

We have confirmed and updated the ecological preferences presented in lines 52-56 (previous 50-54) on the basis of data from other regions of the world, and included appropriate references.

Question 2. The same can be applied to the sentence in lines 66-67.

This sentence concerns Poland, in lines 77-78 (previous 66-67) we added the appropriate word.

Question 3. Similarly, I think that a general description of the distribution of this species worldwide would be desirable in lines 71-76. I would recommend the authors to check the distribution of the species in the Natura 2000 net at: https://natura2000.eea.europa.eu/#

We have added a general description of the distribution of this species in the world. We insert this fragment in our opinion, in a more adequate place, i.e. in line 70 et seq. We thank you for the hint and link. We have re-edited the description presented in lines 83-89 (previous 71-76).

Question 4. In the same line, I think that the discussion and conclusions could be improved if they are slightly re-written with a more general scope. I think that some of the results of the authors are of general interest to characterize and understand the real distribution of this moss (such as their findings on new types of wood), but this can only be addressed if the current discussion is completed with the knowledge on this moss worldwide. I would like to point out one final concern regarding the conclusions of this paper. There are some parts of the manuscript (such as lines 325-326 or lines 332-337) where the authors suggest that according to their results this moss could be deleted from the lists of protected species. I think that the authors should be more careful on this statement.

The Natura 2000 initiative aims to protect habitats that are endangered or rare at a European scale. Buxbaumia viridis distribution is linked to fir forests (as the authors confirm for example in lines 363-366), which is an undoubtedly rare type of forest when the whole Europe is considered. So, in the same line as my previous suggestions, I would avoid to conclude that B. viridis should not be included in protection lists at the European scale when its distribution is not studied at this global scale. Even though Poland and the surrounding territories can provide suitable environments that favour the development of this species, the authors cannot forget the larger scope and aim of the Nature 2000 initiative.

Answer 4. In our opinion, the statements contained in lines 369-370 (previous 325-326) are balanced and are confirmed by the population size in Natura 2000 areas to which the Reviewer provided a link. According to SDF data, e.g. in Natura 2000 areas GR1340009, GR1340003, GR1340010, the population size of B. viridis was from 180 to 1000 individuals, and in the Natura 2000 area ROSCI0194 from 2000 to 5000 individuals.

In lines 377-383 (previous 332-337), before making a decision to change the conservation status of a species, we point out the need to conduct research on the abundance of this species in other regions of Poland and Europe, similar to that in the Bieszczady Mountains.

Literature data show that B. viridis prefers dead fir [1] and spruce wood [2-3]. There are no monoculture spruce stands in the Bieszczady Mountains and the share of spruce in the species composition is very small, which was noted in lines 416-418 (previous 366-368). Beech or fir stands dominate here. In lines 413-416 (previous 363-366) we wrote that due to these habitat preferences, stands with a large share of fir were selected for the search. The point was that we did not search in beech stands or with a predominant share of beech. In many regions of Europe and Poland, B. viridis occurs in spruce stands. Spruce stands are not a rare type of forest in Europe.

  1. Holá, E.; Vrba, J.; Linhartová, R.; Novozámská, E.; Zmrhalová, M.; Plášek, V.; Kučera, J. Thirteen years on the hunt for Buxbaumia viridis in the Czech Republic: still on the tip of the iceberg? Acta Soc. Bot. Pol. 2014, 83, 137–145, doi:10.5586/asbp.2014.015.
  2. Taylor, S. Buxbaumia viridis in Abernethy Forest and other sites in northern Scotland. Bryol. 2010, 100, 9–14.
  3. Wiklund, K. Substratum preference, spore output and temporal variation in sporophyte production of the epixylic moss Buxbaumia viridis. Bryol. 2002, 24, 187–195, doi:10.1179/037366802125001358.

Minor mistakes:

Question 5. Please check the “An additional 17 localities” (lines 204-205)

Answer 5. We check this expression, it’s OK.

Question 6. Figure 2 legend (lines 228-229): I would change the explanation of “c” to something like "surveyed areas without viridis findings"

Answer 6. We change the explanation of “c” to "surveyed areas without B. viridis findings".

Question 7. Tables 2, 4 and 5 legends: please, explain what do “a” and “b” superscripts mean. Current expression is not clear. Which is the difference on significance among the two letters?

Answer 7. We have re-edited the explanations under Table 2 lines 278-283 (previous 245-248) and table 4 line 308-311. The legend of table 5 remained unchanged. The present explanation is a typical expression used to describe the results of such statistical analysis. If the values of the variable do not differ statistically, they have the same symbol (letter) in the index, and if they differ, they have a different symbol (letter).

Question 8. Figure 4 legend: this is not clear, which is the difference between a and b?

Answer 8. We have re-edited the explanations under Figure 4. The answer to the question is included above (see answer 7).

Question 9. Table 7 legend: same comment as above. And be aware that there is only letter “a” here, the legend cannot say “different letters”

Answer 9. As suggested by the Reviewer 2, we changed the legend to table 7.

Question 10. Discussion: please delete lines 308-3011

Answer 10. We delete lines 351-354 (previous 308-311).

Question 12. Line 437, use italics for viridis

Answer 11. We use italic for B. viridis.

Question 13. Conclusions: please delete lines 458-459

Answer 11. We delete lines 514-515 (previous 458-459).
